# The vertical structure of precipitation at two stations in East Antarctica derived from micro rain radars

Claudio Durán-Alarcón[1], Brice Boudevillain[1], Christophe Genthon[1], Jacopo Grazioli[2,3],
Niels Souverijns[4], Nicole P.M. van Lipzig[4], Irina V. Gorodetskaya[5], and Alexis Berne[2]

[1]Université Grenoble Alpes, CNRS, IRD, Grenoble INP institute of engineering, IGE, 38000 Grenoble, France
[2]École polytechnique fédérale de Lausanne, Environmental Remote Sensing Laboratory, CH-1015 Lausanne, Switzerland
[3]Federal Office of Meteorology and Climatology, MeteoSwiss, Locarno-Monti, Switzerland
[4]Katholieke Universiteit Leuven, Department of Earth and Environmental Sciences, Leuven,Belgium
[5]University of Aveiro, Department of Physics, Centre for Environmental and Marine Studies, Portugal

**Correspondence:** Brice Boudevillain (brice.boudevillain@univ-grenoble-alpes.fr)

**Abstract.**

Precipitation over Antarctica is the main term in the surface mass balance of the Antarctic ice sheet, which is crucial for the future evolution of the sea level worldwide. Precipitation, however, remains poorly documented and understood mainly because of a lack of observations in this extreme environment. Two observatories dedicated to precipitation have been set up at the Belgian station Princess Elisabeth (PE) and at the French station Dumont d'Urville (DDU) in East Antarctica. Among other instruments, both sites have a vertically-pointing micro rain radar (MRR) working at the K-band. Measurements are continuously collected at DDU since the austral summer 2015-2016, while they have been collected mostly during summer seasons at PE since 2010, with a full year of observation during 2012. In this study, the statistics of the vertical profiles of reflectivity, vertical velocity and spectral width are analyzed for all seasons. Vertical profiles were separated into surface precipitation and virga to evaluate the impact of virga on the structure of the vertical profiles. The climatology of the study area plays an important role in the structure of the precipitation: warmer and moister atmospheric conditions at DDU favor the occurrence of more intense precipitation compared with PE, with a difference in 8 dBZ between both stations. The strong katabatic winds blowing at DDU induce a decrease of reflectivity close to the ground due to the sublimation of the snowfall particles. The vertical profiles of precipitation velocity show significant differences between the two stations. In general, at DDU the vertical velocity increases as the height decreases, while at PE the vertical velocity decreases as the height decreases. These features of the vertical profiles of reflectivity and vertical velocity could be explained by the more frequent occurrence of aggregation and riming at DDU compared to PE, because of the lower temperature and relative humidity at the latter, located further in the interior. Robust and reliable statistics about the vertical profile of precipitation in Antarctica, as derived from micro rain radars for instance, are necessary and valuable for the evaluation of precipitation estimates derived from satellite measurements and from numerical atmospheric models.

# 1 Introduction

Solid precipitation is a key component of the hydrological cycle in high-altitude and high-latitude regions. In Antarctica, precipitation falls mainly in form of snow and plays an important role as the largest positive term in the surface mass balance (SMB) of the Antarctic ice sheet (van Wessem et al., 2014, 2018). Climatological variations in precipitation regime can therefore significantly affect the SMB and thus the global sea level (Krinner et al., 2007; Mengel et al., 2016). Under different scenarios of climate warming, precipitation in the Antarctic region is expected to increase, due to an increase in the atmospheric moisture-holding capacity (Ligtenberg et al., 2013; Frieler et al., 2015; Tang et al., 2018). According to Palerme et al. (2017), most of the models involved in the Fifth Climate Model Intercomparison Project (CMIP5) agree on the increase of precipitation, with an average change between 6 and 25% by the end of the 21$^{st}$ century, depending on the warming scenario. Nevertheless, an evaluation of the capacity of the models to simulate the current precipitation in Antarctica, using Cloudsat products as reference, reveals that most of the models overestimate the mean annual precipitation rate, reaching errors higher than 100% in some cases (Palerme et al., 2017; Lemonnier et al., 2018). These results pose an important challenge in improving the current modelling of precipitation in Antarctica, and thereby having more confidence in the projections.

Although local net accumulation is often used as proxy for snowfall (e.g. Frezzotti et al., 2004), it is largely affected by precipitation conditions upstream of the site, leading to wind-driven snow transport to (or away from) the site (e.g. Van den Broeke et al., 2004; Souverijns et al., 2018a). Ground-based radar instruments provide suitable information to monitor vertical variations of precipitation, through the collection of range-resolved Doppler radar observations (e.g. vertical profile of reflectivity). The study of the vertical structure of precipitation is fundamental to understand the dynamical and microphysical processes controlling hydrometeors formation and evolution toward the surface, as well as to evaluate numerical atmospheric models and satellite precipitation products. Spaceborne active sensors, such as the Cloud Profiling Radar (CPR) on board of Cloudsat (and next on EarthCare), have the potential to monitor precipitation in a large horizontal and vertical extent in Antarctica (Palerme et al., 2014). However, this type of data source presents a generalized lack of observation close to the surface level, due to ground-clutter contamination, producing the so-called "blind zone" over the surface (the lower 1.2 km over land or ice for the case of Cloudsat) (Maahn et al., 2014). In this context, ground-based vertical profilers provide an advantage in terms of more detailed monitoring near the surface.

Significant efforts have been invested to monitor and better understand precipitation at two locations: the project APRES3 (Antarctic Precipitation Remote Sensing from Surface, http://apres3.osug.fr, Grazioli et al. (2017a)) at the French Dumont d'Urville station on the coast of Adélie Land and the project HYDRANT (HYDRological cycle in ANTarctica, https://ees.kuleuven.be/hydrant/hydrant.html, Gorodetskaya et al. (2015)) and its follow up AEROCLOUD (How do AEROsols and CLOUDs affect the East Antarctic climate?, https://ees.kuleuven.be/hydrant/aerocloud/) at the Belgian Princess Elisabeth station in Dronning Maud Land, both of them implemented with a vertical-pointing K-band micro rain radar (MRR). Other recent initiatives that also study precipitation in Antarctica are the micro rain radar observations collected at the Italian station Mario Zucchelli (Souverijns et al., 2018b) and the AWARE (ARM West Antarctic Radiation Experiment) field campaign organized by

ARM/ASR (Atmospheric Radiation Measurement/Atmospheric System Research) which involved multiple radars at various frequencies and satellite-based remote sensing observations at McMurdo station (Lubin et al., 2017).

The main objective of this work is to characterize the vertical structure of the precipitation from profiles of the Doppler moments from MRRs located at these two different sites in East Antarctica, analyzing the vertical structure of precipitation throughout the year to understand the main microphysical processes involved in its variability. This knowledge will represent a significant input in the calibration and validation of satellite observation in Antarctica and modeling purposes. This manuscript is structured as follows: section 2 provides a description of the study area, the data and methodology used in this study; section 3 presents the overall statistics of the vertical profiles; section 4 analyses the importance of surface precipitation and virga in the study area; section 5 is the seasonal analysis and section 6 delivers a summary and the main conclusions of the work.

## 2    Material and methods

### 2.1    Study area

Two different sites are studied in East Antarctica, corresponding to the APRES3 and HYDRANT/AEROCLOUD observatories, located in Dumont d'Urville (DDU) and Princess Elisabeth (PE) stations, respectively.

DDU is located on the Petrels Island (66°39'S, 140°00'E) at 41 m a.s.l. at the coast of Adélie Land. One of the strongest and most directional katabatic regime dominates this region with an annual average wind speed of 10 m s$^{-1}$ (König-Langlo et al., 1998; Bromwich et al., 2011; Grazioli et al., 2017a) and an annual precipitation rate of about 679 mm w.e. yr$^{-1}$ (liquid water equivalent per year) (Palerme et al., 2014). Katabatic winds coming from the interior of the Antarctic continent are responsible for the presence of significant blowing snow events and for the sublimation of a significant part of this blowing snow and precipitation, reducing the total amount of snow at ground level (Grazioli et al., 2017b).

PE is located in Dronning Maud Land (71°57'S, 23°21'E), 173 km inland and at 1392 m a.s.l. on the Utsteinen Ridge, in the escarpment zone at the north of the Sør Rondane mountains range (Gorodetskaya et al., 2015). Meteorology at PE is characterized by alternation of two regimes, a cold katabatic regime with low wind speeds and humidity, strong near-surface temperature inversion and high surface pressure and a warm synoptic regime with strong wind speeds, high specific humidity and low surface pressure (Gorodetskaya et al., 2013). Snowfall is generally associated with a cyclone located north-west or north of the station (Souverijns et al., 2018a). Unlike at DDU, katabatic winds at PE are mostly attenuated by the blocking effect of the mountain range, but still heavy blowing snow events (reaching up to 30 m height) occur 13% of the time, mostly during transitional periods caused by strong synoptic winds (Gossart et al., 2017). Based on a full year of MRR observation at PE in 2012, the total annual precipitation ranges from 87 to 266 mm w.e. yr$^{-1}$, according to uncertainty in Z-S relationship as described by Souverijns et al. (2017).

Figure 1 displays the location of both stations on a elevation map.

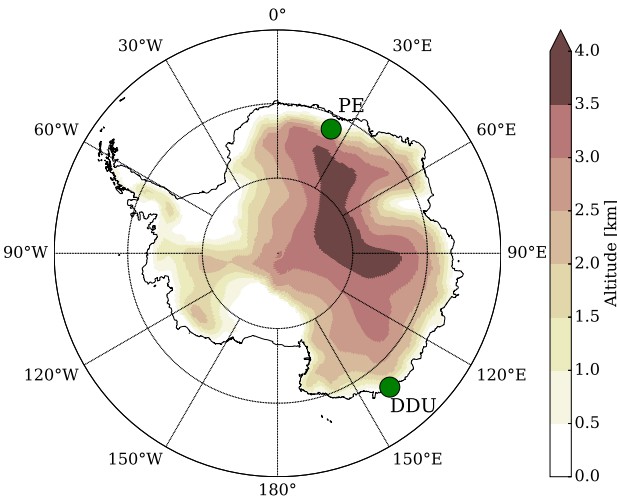

**Figure 1.** Topography (Bamber et al., 2009) of the Antarctic Continent. Green points represent the location of Dumont d'Urville (DDU) and Princess Elisabeth (PE) stations.

## 2.2 Ground-based MRR observations

APRES3 and HYDRANT/AEROCLOUD observatories are both equipped with a K-band vertically pointing MRR-2 manufactured by METEK, deployed with the aim of long-term monitoring of precipitation at DDU and PE respectively. Measurements at DDU are continuously collected since November 2015 (Grazioli et al., 2017a), while at PE in-situ observations are carried

out mainly during summer campaigns between 2010 and 2016, including one full year of measurements in 2012 (Gorodetskaya et al., 2015; Souverijns et al., 2017). In this study, two full years of data at DDU and all the observations at PE are used for the analysis.

MRR is a frequency-modulated continuous-wave (FMCW) system with low requirement of energy (small transmitter power: 50 mW) and of supervision during operation time (Peters et al., 2005), which makes this instrument suitable for monitoring

remote locations with minimum logistic support (Gorodetskaya et al., 2015; Grazioli et al., 2017a).

Both radar systems use the same vertical (100 m) and temporal (1 min) resolutions, as well as the same number of range gates (31 up to 3 km) and of Doppler velocity intervals (64 between 0 and 12 m s$^{-1}$). More details of the configuration of MRR and the complemented instrumentation for DDU are described by Grazioli et al. (2017a) and Genthon et al. (2018), and for PE by Gorodetskaya et al. (2015) and Souverijns et al. (2018b). The MRR at DDU is deployed inside a radome to protect

the instrument from the inclement Antarctic conditions in Adélie Land, and the MRR at PE has no radome since the weather conditions are not as harsh as at DDU. The effect of the radome on the MRR at DDU was evaluated and corrected by Grazioli et al. (2017a) using a co-located X-band MXPol radar system. The comparison of both radar reflectivity observations showed that the radome produces an average attenuation of 6 dB in the data. A more recent analysis, using a second MRR at DDU deployed outside of the radome, confirms this estimate of the radome attenuation.

## 2.3 MRR post-processing

Originally, the MRR was developed as a rainfall profiler, exploiting an idealized relationship between size and fall velocity of liquid precipitation (Atlas et al., 1973) and assuming absence of vertical wind (Peters et al., 2002, 2005). For the case of snowfall, the large variability of particle shapes and densities changes the size-fall velocity relationship and introduces a large uncertainty in the estimation of the snow rate, thus this approach is not suitable for snow precipitation. Recently, new approaches have been proposed to address this problem through the direct integration of the Doppler spectrum to calculate Doppler radar moments such as effective reflectivity (Ze), mean Doppler velocity (W) and spectral width ($\sigma$) (Kneifel et al., 2011; Maahn and Kollias, 2012). In this work, MRR data were processed using the method proposed by Maahn and Kollias (2012) (hereafter noted MK12) that improves the noise filtering algorithm and implements a dynamic procedure to dealiase the Doppler spectrum, allowing to take into account very small and negative W (cases of weak updraft). Despite of the Doppler velocity correction, turbulence still presents a problem for the dealiasing procedure and can present a problem for low level observations, especially for the Doppler velocity. The two lower ranges are usually not considered in the post-processing because they can be strongly affected by the near-field effect and the two highest range gates also are excluded from the analysis due to the high noise in the signal (MK12). After post-processing, the sensitivity of MRR ranges between -14 and -8 dBZ depending of the height level (Maahn and Kollias, 2012). At DDU, the attenuation due to the radome (estimated at about 6 dBZ, see Grazioli et al. (2017a)) must however be taken into account and leads to a lower sensitivity.

The instruments exhibited different types of noise at the two stations that were not completely filtered out after the post-processing proposed by MK12. For MRR at DDU during given periods, a spread noise that covers all the range bins, especially during clear sky conditions, was present. For MRR at PE (and also a second MRR deployed at DDU) sometimes a particular artifact between different range gates was observed, encompassing from a few minutes up to a full day.

MRR at DDU sometimes experienced interruptions during the acquisition time, leading to a decrease in the number of observations per minute, collecting 2 instead of ~6 profiles per minute. This decrease of the sampling rate leads to an increase of the normalized standard deviation ($V_T$) of a set of spectra and MK12 interpreted the signal as a container of potential peak due to precipitation. Therefore, the $V_T$ threshold used by MK12 to remove clear-sky profiles from the post-processing was adapted to be dependent on the sampling rate of the instrument (See Appendix A). All clear sky noise was removed from MRR data.

For MRR at PE, the presence of artifacts could be due to low-frequency interferences produced by nearby electronic devices. The affected range gate depends on the frequency of the interference and adjacent gates are affected by leakage between gates. The interference produces a strong peak in the Doppler spectrum which frequently appears in the largest velocity gates. 3% of the days at PE containing bands of noise were excluded from the analysis.

## 2.4 Radio soundings

Daily radio soundings are carried out permanently at 00 UTC at DDU station by MeteoFrance since 1956, while at PE only summer radio soundings are available since 2014, collected at 12 UTC by the Royal Meteorological Institute of Belgium. In

this work, we use vertical profiles of air temperature and relative humidity to characterize different precipitation types and seasons (only summer at PE), corresponding to simultaneous observations of radio soundings and MRR. Relative humidity ($RH$) with respect to liquid water is converted into relative humidity with respect to ice ($RHi$) using the ratio of the saturation vapor pressure over water $e_{sw}$ to the saturation vapor pressure over ice $e_{si}$ as it is shown in the following equation:

$$RHi = RH \cdot \frac{e_{sw}}{e_{si}}, \tag{1}$$

$e_{sw}$ and $e_{si}$ are derived using the equations of Goff (1957), detailed in Appendix B.

## 2.5 Statistics of vertical profiles and temporal integration

The statistics of the vertical profiles of reflectivity (VPR), of vertical velocity (VPV) and spectral width (VPS) are analyzed at DDU and PE. Considering the high variability of short time observations influenced by advection and turbulence, different temporal integration intervals are evaluated in order to choose an optimal integration time for analysis. The variability of the average VPR at fixed time intervals between 1 min and 12 h is analyzed at both stations. Average VPR are calculated in linear units ($mm^6 \, m^{-3}$) as the sum of the 1-min VPRs along fixed windows of $t$ duration, divided by $t$ and then converted in dBZ as in Welsh et al. (2016).

Figure 2a and c show the average VPR at different temporal resolutions for DDU and PE respectively, compared to the original 1-min resolution. Significant differences can be observed between the absolute values of VPR between the two stations, with a higher Ze at DDU than at PE and greater variability between integration intervals. These differences in the average VPR at the two stations, are the first indication about the presence of different growth habits and size distribution of ice particles at DDU and PE. Ice growing procesess, such as riming and aggregation, that depend on temperature and the moisture content in the atmosphere, can play an important role in the increase of Ze. The relatively warmer conditions at DDU and the proximity to the a cyclogenesis region (Bromwich et al., 2011), may explain the observed differences with respect to PE.

Variations of VPR between temporal resolutions $t$ are compared using the following expression:

$$\delta = \frac{|\Delta Ze|}{\Delta t \cdot Ze} \tag{2}$$

where $\Delta Ze$ is the absolute difference in Ze between consecutive temporal integration, $Ze$ is the average VPR and $\Delta t$ is the increase in time resolution, with a constant value equal to 15 min, except for the first interval between 1-min and 15-min temporal integration steps. The values of VPR decrease as the temporal integration increases, but at each step the Ze values tend to decrease more slowly. Figure 2, panels b and d show the mean, minimum and maximum variation ($\delta$) of the average VPRs with respect to the time integration steps, for DDU and PE respectively. At both stations, VPR presents large changes between shorter time scales and it is more stable for larger time resolution. Maximum and minimum variation in the vertical profile also decrease, which means that the shape of the VPR also becomes more stable. The integration times of 1 h and higher represent more stationary patterns in the vertical profile than short time integration, that can be affected by advection or turbulence. Based on this analysis, 1 h was considered as an optimal temporal integration to study climatological patterns in the VPR, removing the short time perturbations while keeping enough observations to make a robust statistical analysis.

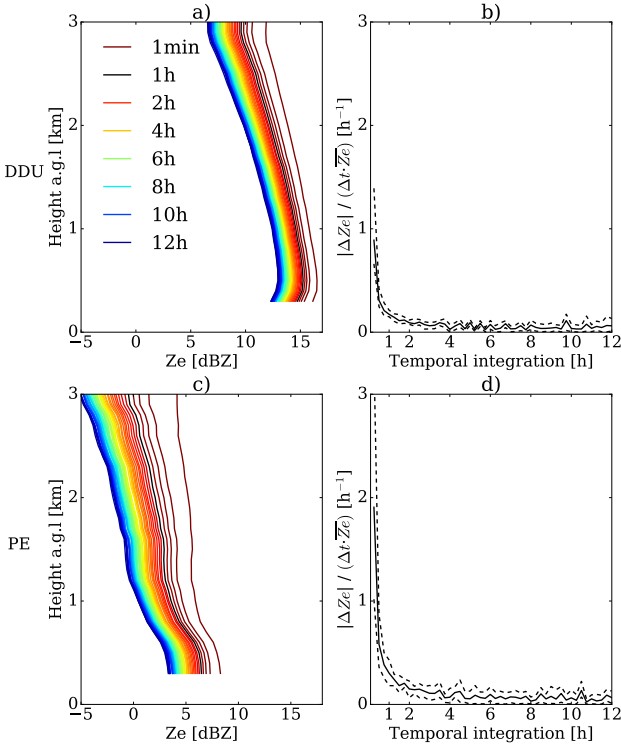

**Figure 2.** Average VPR at different temporal integrations are displayed in (a) and (c) panels, for DDU and PE respectively. Colored lines represent different temporal integration steps, corresponding to 1 min and from 15 min to 12 hours separated by 15-min intervals (legend displays only a few selected time steps and 1-h VPR is represented in black). The temporal variability between time steps is shown in panels (b) and (d), for DDU and PE. Bold lines correspond to the mean reduction of Ze in function of the temporal integration step, weighted by time interval and the mean Ze at the given time scale. Dashed lines are the maximum and minimum reductions of Ze within the whole average VPR.

After the selection of the temporal integration, VPV and VPS are averaged at the same temporal resolution. In these cases, the average does not consider the zero values (clear sky situations), as the case of VPR, because $W$ and $\sigma$ are considered as intensive properties of precipitation, which means that they are independent of the amount of precipitation within a given temporal interval.

## 2.6 Precipitation profile classification

We classified the vertical profiles of precipitation in two different categories based on whether the precipitation reaches the surface or not. Snowfall sublimation depends on both the properties of the solid hydrometeors (e.g. bulk density, terminal velocity and particle size distribution) and meteorological conditions (e.g. relative humidity, temperature) (Clough and Franks, 1991; Maahn et al., 2014). Precipitation sublimation and virga may occur often in Antarctica under temperature inversion and unsaturated air conditions (Maahn et al., 2014; Grazioli et al., 2017b). At both sites, virga have been reported in previous works

by using ground-based remote sensing techniques, such as radar and lidar instruments (e.g. Gorodetskaya et al., 2015; Maahn et al., 2014; Grazioli et al., 2017a).

MRR is especially suitable to detect hydrometeors that have reached a sufficient size to fall as precipitation, including streaks of virga, but it is insensitive to cloud particles (Gorodetskaya et al., 2015; Souverijns et al., 2017). For this reason, the presence of precipitation signal in the lowest reliable MRR range gate of each profile was used to separate virga observations from surface precipitation. Occurrence of echoes at 300 m height was used as reference of surface precipitation since there is no useful signal below this height for all the profiles at both stations.

## 3   Overall statistics

The statistics of all the MRR profiles for the complete observation period at both stations were carried out, prior to the classification of the precipitation profiles and the seasonal analysis, to obtain a general picture of the distribution of the variables of interest. A total of 5331 and 5058 hourly vertical profiles were obtained at DDU and PE during all respective data periods. Figure 3 shows the distribution of Ze, W and $\sigma$ in the vertical profile extension in percentage, normalized by the total number of the range gates containing precipitation. In the following subsections, the statistics for vertical profiles of the three Doppler radar moments are presented.

### 3.1   Vertical profiles of reflectivity

At both stations, mean and median VPR describe a general increase from the top toward the surface. Variability of Ze depends on the microphysical and scattering properties of the targets such as the particle size distribution (PSD) and the complex refractive index ($\underline{n}$). Previous studies have shown that in the ice part of precipitating clouds and snow storms, the ice growth by vapor deposition, riming and/or aggregation have an important role on the vertical evolution of PSD and thus on the vertical patterns of radar VPR, which may cause the observed increase of Ze toward the surface (Moisseev et al., 2009, 2015; Bechini et al., 2013; Schneebeli et al., 2013; Pfitzenmaier et al., 2018). Considering that dry snow is a mixture of ice and air, the dielectric properties of a particle depend on the proportion of ice and air that it contains, thus $\underline{n}$ is sensitive to the snow/ice particle type and bulk density (Tiuri et al., 1984; Sadiku, 1985). The vertical profiles of snow types have been identified with the dual-polarization weather radar observations collected at DDU by Grazioli et al. (2017a), who found that more pristine particles (e.g. dendrites, columns) are largely dominant above 2.5 km of height and that the proportion of aggregates and rimed particles significantly increases with decreasing height. For the case of PE, rimed particles and graupel are less frequent due to the colder temperatures in this region (Souverijns et al., 2017).

Mean VPR at DDU experiences a peak of maximum Ze of 15.3 dBZ close to the ground followed by a decrease toward the surface to 14.7 dBZ. This particular pattern is associated with an enhanced sublimation process driven by low-level katabatic winds leading to unsaturated air conditions that favor the sublimation of ice particles, especially associated with lighter snowfall (Grazioli et al., 2017b), but it is not clearly observed at PE (see Figure 3a, b).

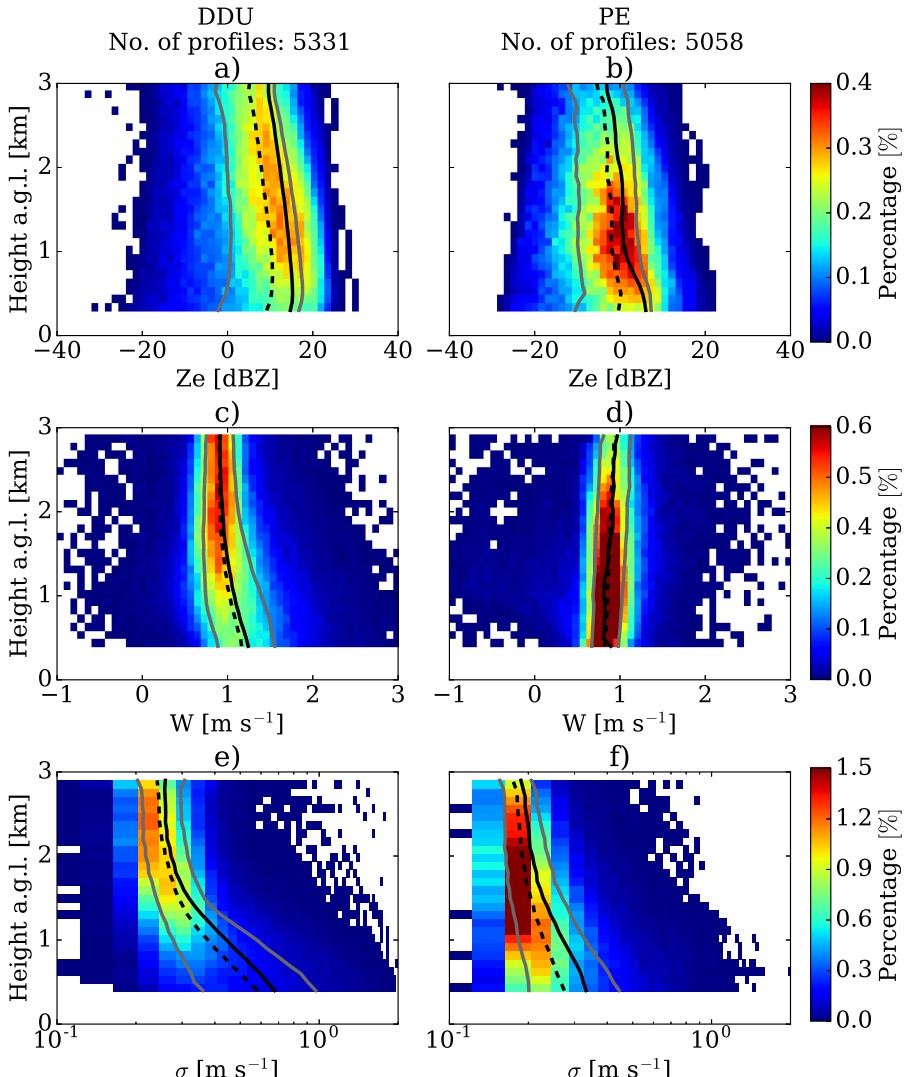

**Figure 3.** Frequency by altitude diagram for Ze (a, b), W (c, d) and $\sigma$ (e, f) values observed at DDU (a, c, e) and PE (b, d, f). Solid-black and dashed-black lines represent the average and median vertical profiles of Ze, W and $\sigma$ respectively. Grey lines correspond to the 20 and 80% quantiles of the vertical profiles. $\sigma$ values are plotted in log x-axis to highlight the variations for small values.

At PE, where katabatic winds are less strong than at DDU due to the Sør Rondane Mountain sheltering (Gorodetskaya et al., 2013, 2015; Thiery et al., 2012), the mean VPR does not exhibit a decrease close to the surface as it is observed at DDU. It must be noted that most observations at PE were collected in summer when katabatic winds are even weaker. The Ze value at the lowest height is 6.1 dBZ, which corresponds to a difference of 8.6 dBZ compared to DDU at the same range bin.

5      The average VPRs at both stations are larger than the median VPRs at all the range bins, because the mean values are strongly influenced by a few high values. Median and quantiles are not affected by the differences of Ze values and they can

represent the patterns of lower reflectivity values. For instance, low level sublimation is observed on these lower quantiles of VPR (e.g. Q20%) inducing an amplified decrease of Ze near the surface at DDU and even observable at PE, probably because small ice particles within these events are more susceptible to sublimation (Clough and Franks, 1991; Grazioli et al., 2017b).

## 3.2 Vertical velocity profiles

The vertical structure of W presents two different patterns at DDU and PE (see Figure 3c, d). In the case of DDU, mean and median VPV remain constant at about 0.9 m s$^{-1}$ from 3 to 1 km of height, and increase up to 1.2 m s$^{-1}$ in the lowest km above ground. At PE, a limited decrease from about 1 m s$^{-1}$ at 3 km of height to about 0.9 m s$^{-1}$ at 400 m of height is observed.

   W represents the mean of the reflectivity-weighted terminal velocities of the scatters, but also is influenced by the vertical air motion. Depending on the type of ice particles, the terminal velocity $V_t$ can be parameterized as function of the maximum size

$D$, based on the different relationships between air friction and Reynolds number ($Re$), which both depend on the air density $\rho_a$ (Böhm, 1989; Mitchell and Heymsfield, 2005; Heymsfield and Westbrook, 2010; Molthan et al., 2010). At DDU, the increase in occurrence of larger hydrometeors towards the ground, mainly of aggregates and to a lesser extent rimed particles (Grazioli et al., 2017a), can explain the observed increase in the VPV in the lowest km above ground.

   At PE, the habits of hydrometeors at the ground are represented by mostly small dendrites, columns, capped columns and

rosettes (Gorodetskaya et al., 2015; Souverijns et al., 2017), that are more affected by air friction, resulting in lower values of W than at DDU. Moreover, for small crystals with low growth along the downward trajectory, the increase of air density can lead to an increase of the air friction and thus to a decrease of $V_t$ and W, which can be the reason for the slight but regular decrease in the VPV toward the surface observed at PE. Previous works have used a power-law relationship between size and terminal velocity with an air density correction factor that take into account the inverse relation between $V_t$ and $\rho_a$ (e.g. Molthan et al.,

2010; Heymsfield et al., 2013):

$$V_t(D) = aD^b \left( \frac{\rho_0}{\rho_a} \right)^w \tag{3}$$

where $\rho_0$ is the air density at the surface, $a$ and $b$ are the parameters of the power-law relationship depending on the type of ice particle and $w$ controls the correction factor, usually equal to 0.4. According to standard atmospheric conditions, air density can increase from 0.79 to 1.07 kg m$^{-3}$, from the 3 km of height level to the ground at PE, leading to a decrease 11% of terminal

velocity, that is in accordance with the results obtained at PE. Finally, a decrease toward the ground of the fall speed of the crystals can also be expected when air temperatures increase, since air viscosity also increases with temperature (Westbrook, 2008). A more detailed comparison of VPS in the common range of altitudes above sea level at the two stations is presented in Appendix C.

## 3.3 Vertical profiles of spectral width

For a vertically pointing radar, the spectral width ($\sigma$) represents the variability of the vertical velocity due to the variety of particle sizes and shapes within the sampling volume (Garrett et al., 2015; Maahn and Kollias, 2012). The variability of the vertical velocity is also influenced by the environmental turbulence (i.e. strong turbulence leads to a broad spectrum) (Nastrom,

1997; Garrett and Yuter, 2014). In the case of DDU, the average VPS describes no significant variations 2 and 3 km a.g.l., but it exhibits a constant increase toward the surface along the 2 km near the ground. This increasing of VPS occurs at the same height where VPV begins to increase and also where the differences between Q20% and Q80% of W increases (see Figure 3e). One factor that can explain this increase of VPS toward the surface is the increase of rimed particles and aggregates that co-exist with small ice particles, generating spectra with low and high Doppler speeds. Moreover, the interaction of the particles with the turbulent katabatic layer below 2 km, can increase the spectral width (Parish et al., 1993).

On the other hand, the VPS at PE also increases toward the surface, but less pronounced than at DDU. Figure 3e shows the probability of occurrence of $\sigma$ at different heights. Dominant presence of small particles, with low probability of rimed or aggregated particles, leads to lower variations of VPS.

## 4   Surface precipitation and virga

Previous studies (e.g. Maahn et al., 2014; Gorodetskaya et al., 2015; Grazioli et al., 2017a, b) and our current observations suggest that virga is a frequent phenomenon in the study area (36% of the profiles at DDU and 47% at PE are virga observations), it is therefore worth to analyze its impact on the climatology of the vertical structure of precipitation in the study area. For that, the statistics of all vertical profiles are analyzed, based on the classification of surface precipitation or virga.

Figures 4 and 5 show the vertical distribution of Ze, W and $\sigma$ values for the entire observation period, separated into surface precipitation and virga, for DDU and PE respectively. The signature of the mean and median VPR of the surface precipitation show important differences with respect to virga. At both locations, VPR of virga has a lower reflectivity than the profiles associated with precipitation reaching the surface, because weaker precipitation is more likely to be completely sublimated. The occurrence of riming/aggregation of the hydrometeors may also play a secondary role in the increase of W toward the surface observed at DDU in virga profiles.

Ice virga occurs when the low troposphere is dry, ice particles are small and do not experience significant growth, which are ideal condition for ice sublimation (Clough and Franks, 1991). In the case of DDU, the peak of the mean VPR of surface precipitation is 15.8 dBZ, whereas the peak of the virga VPR is 7.5 dBZ. The values closest to the ground for surface precipitation and virga are 14.7 dBZ and -12.9 dBZ, evidencing an enhanced sublimation rate towards the surface for virga profiles. With respect to the VPV, the patterns of surface precipitation are similar to those for all profiles together, however VPV of virga shows a significant increase of W within the lower 1000 m toward the surface. This increase of W for virga profiles at DDU suggests that small particles are the first to sublimate completely, leading to an increase of the mean Doppler velocity because of the biggest particles.

At PE, the values of Ze are lower compared to DDU, but they also present significant differences between VPR of surface precipitation and virga. The maximum value of the mean VPR of surface precipitation is 6.8 dBZ and 1.1 dBZ for the case of virga. The respective values near the ground are 6.5 and -13.4 dBZ. The VPV shows a steady but constant decrease of the vertical velocity for both precipitation reaching the surface and virga. The fast increase of W for ice virga does not happen at PE, which can be explained by the lower occurrence of aggregation and riming, leading to lower vertical velocities and narrow

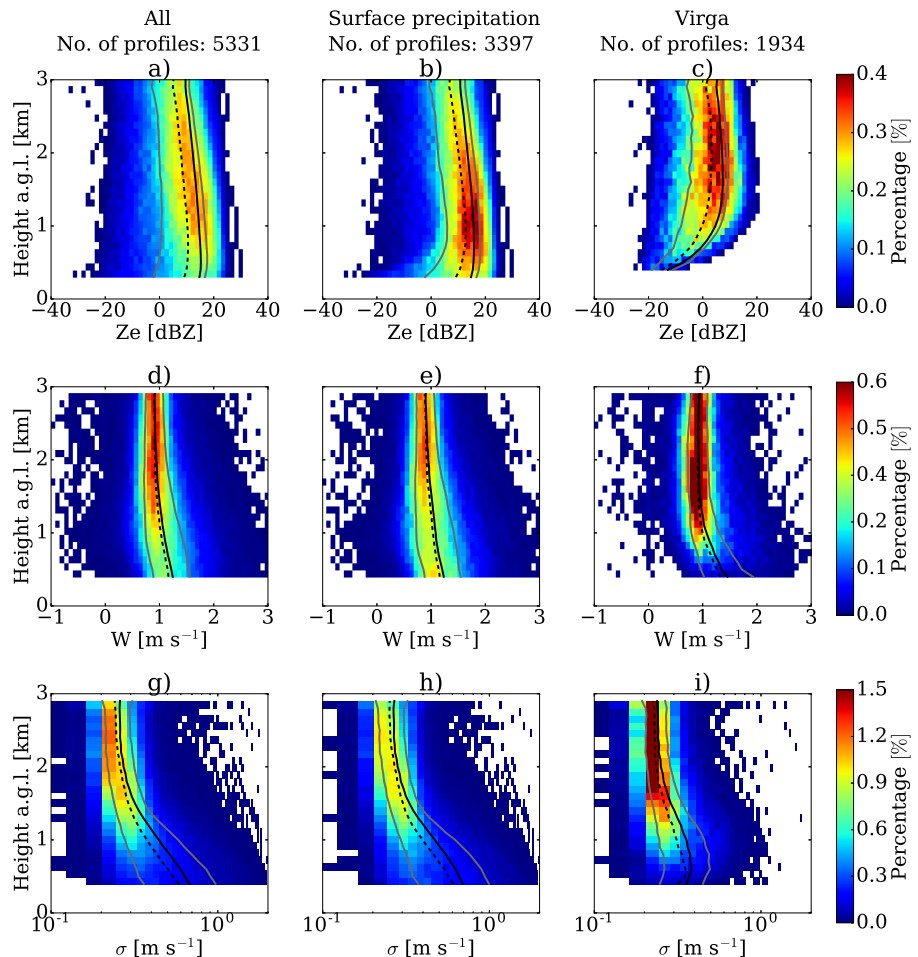

**Figure 4.** Frequency by altitude diagram for Ze (a, b, c), W (d, e, f) and $\sigma$ (g, h, i) values observed at DDU, separated by type of snowfall. (a, d, g) correspond to all the profiles, (b, e, h) to surface precipitation and (c, f, i) to ice virga. Solid-black and dashed-black lines represent the average and median vertical profiles of Ze, W and $\sigma$ respectively. Grey lines correspond to the 20 and 80% quantiles of the vertical profiles. $\sigma$ values are plotted in log x-axis to highlight the variations for small values.

spectral width, compared with DDU. The two lower layers of mean VPV for virga show an irregular shape because the mean value become more sensitive to the extremes when the number of observations is reduced (see Figure 5f). Median VPV does not present this problem at the lowest bin gates and it shows the same decreasing pattern of the rest of the profile.

The VPS presents similar behavior at both stations as visible in Figure 4 and 5 (g, h and i). For surface precipitation, VPS increases toward the surface similarly for all profiles together. For the case of ice virga, VPS increases going to the surface but decreases within the lower 1000 m. Although there is an increase in turbulence that increases the value of $\sigma$, in the last part of the trajectory of the particles, a large part have been sublimated, especially those of small size, reducing the breadth of the velocity spectra.

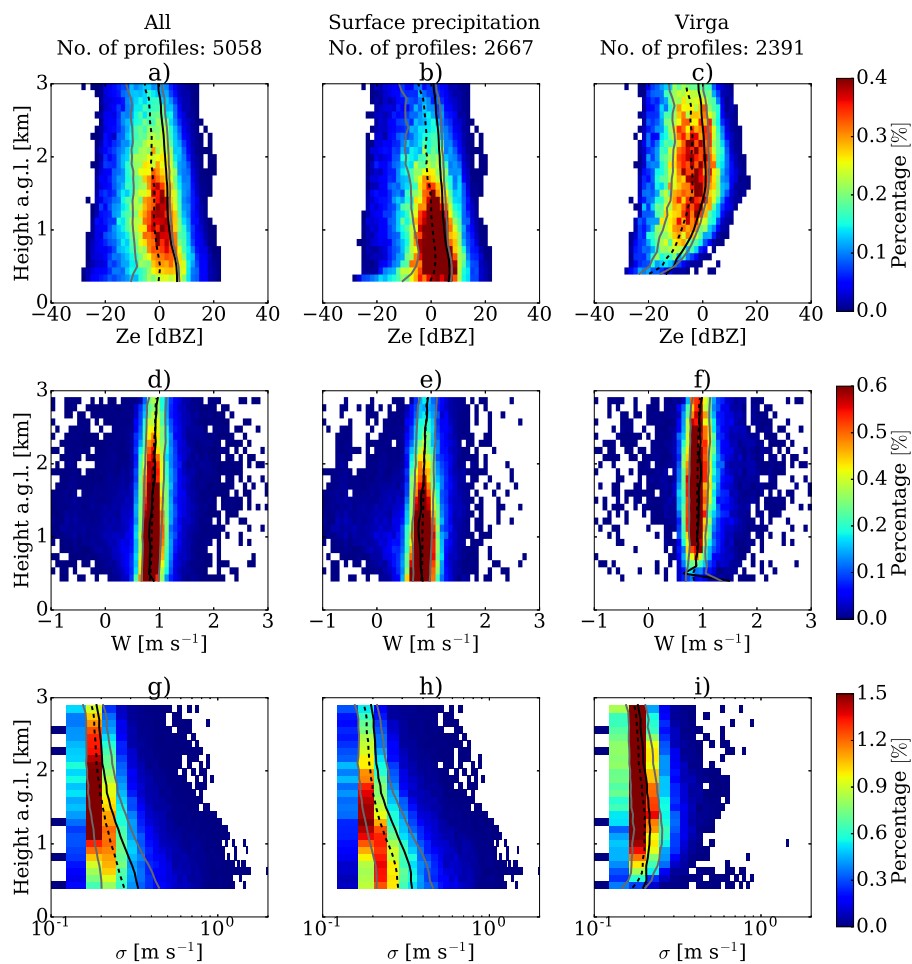

**Figure 5.** Same as Figure 4, but for Princess Elisabeth station.

The low-level sublimation of surface precipitation is more evident when the profiles of virga are removed from the analysis. This can be observed in both stations, comparing the quantiles 20% of the Figure 4a-b and Figure 5a-b. For all the profiles together, the low Ze values of virga profiles, just above the low-level sublimation layer, smooth the strong curvature that is observed when virga is removed.

These results confirm that virga has a significant impact on the patterns of the vertical profiles of Ze, W and $\sigma$, which makes it necessary to differentiate both types of profiles to analyze the structure of the precipitation in the study area. In the following section, a seasonal analysis of the surface precipitation and virga is presented.

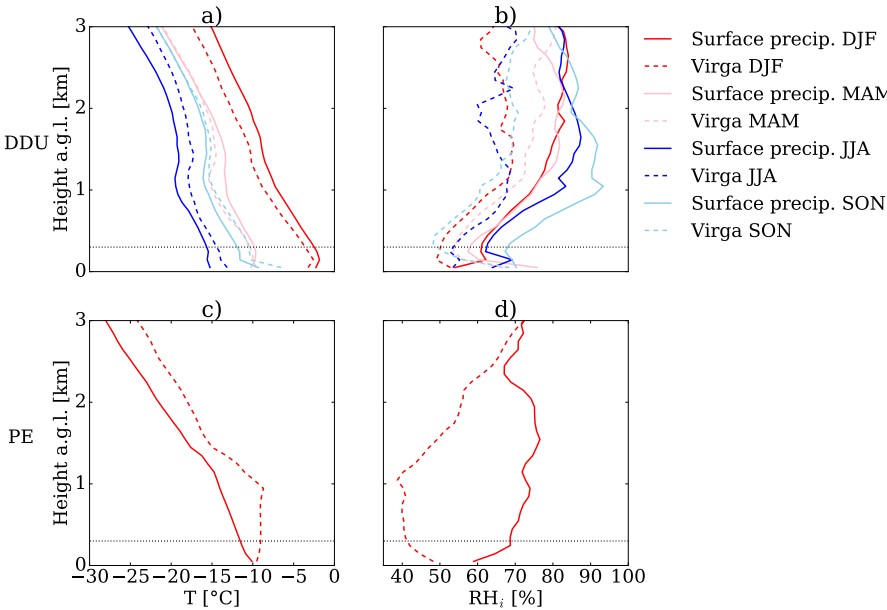

**Figure 6.** Average vertical profiles of air temperature $T$ (a and c) and relative humidity with respect to ice $RHi$ (b and d) during simultaneous time of radio sounding and profiles of precipitation observed with MRR at DDU (a and b) and PE (c and d). Red, pink, blue and sky blue lines represent the summer (DJF), autumn (MAM), winter (JJA) and spring (SON) respectively, and solid lines correspond to surface precipitation, while dashed-lines are virga events. Horizontal doted lines correspond to the height of the lowest available MRR bin (300 m).

**Table 1.** Percentage of surface precipitation and ice virga with respect to the total number of vertical precipitation profiles during all data period and each season at DDU and PE.

| Site | Precipitation type | All seasons | DJF | MAM | JJA | SON |
|------|--------------------|-------------|-----|-----|-----|-----|
| DDU | Surface precipitation | 64 | 56 | 63 | 66 | 69 |
|      | Virga | 36 | 44 | 37 | 34 | 31 |
| PE | Surface precipitation | 53 | 51 | 52 | 63 | 53 |
|      | Virga | 47 | 49 | 48 | 37 | 47 |

## 5 Seasonal variability of vertical profiles

Seasonality of precipitation in Antarctica depends on the availability of moisture in the atmosphere, which depends on the air temperature and the large-scale circulation dynamics in relation to the topography of the continent (Schlosser, 1999; van Lipzig et al., 2002; Marshall, 2009). Coastal region are more affected by the influence of poleward moisture transport, while inland regions are less influenced, because of the blocking by the ice sheet (van Lipzig et al., 2002). Since the seasonality of

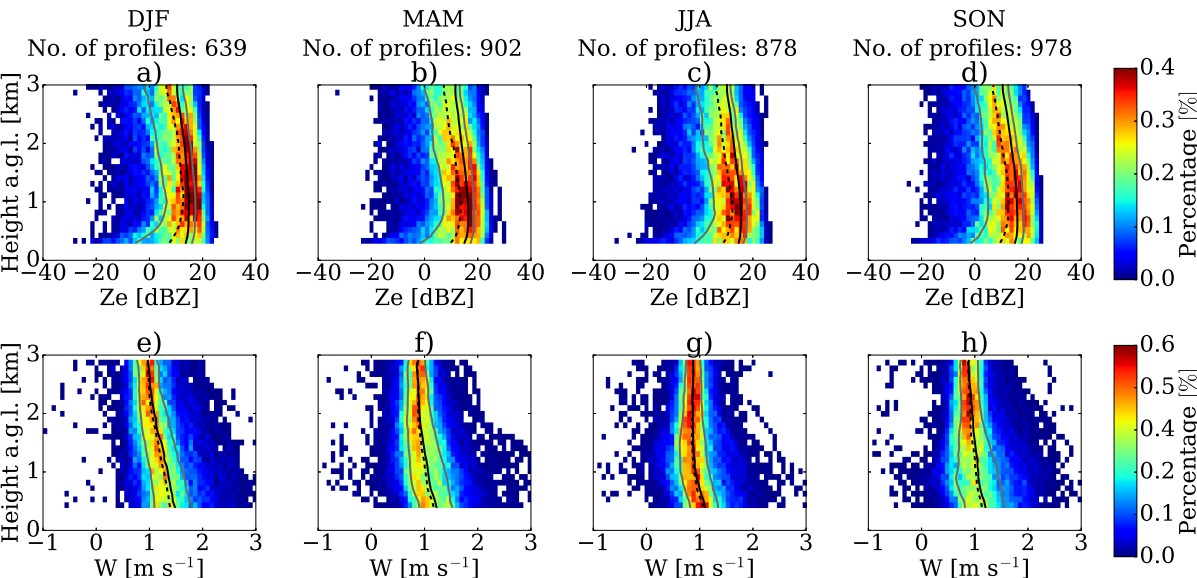

**Figure 7.** Frequency by altitude diagram for Ze (a, b, c) and W (e, f, g) values observed at DDU during DJF (a, e), MAM (b, f), JJA (c, g) and SON (d, h) of surface precipitation. Solid-black and dashed-black lines represent the average and median vertical profiles of Ze and W respectively. Grey lines correspond to the 20 and 80% quantiles of the vertical profiles.

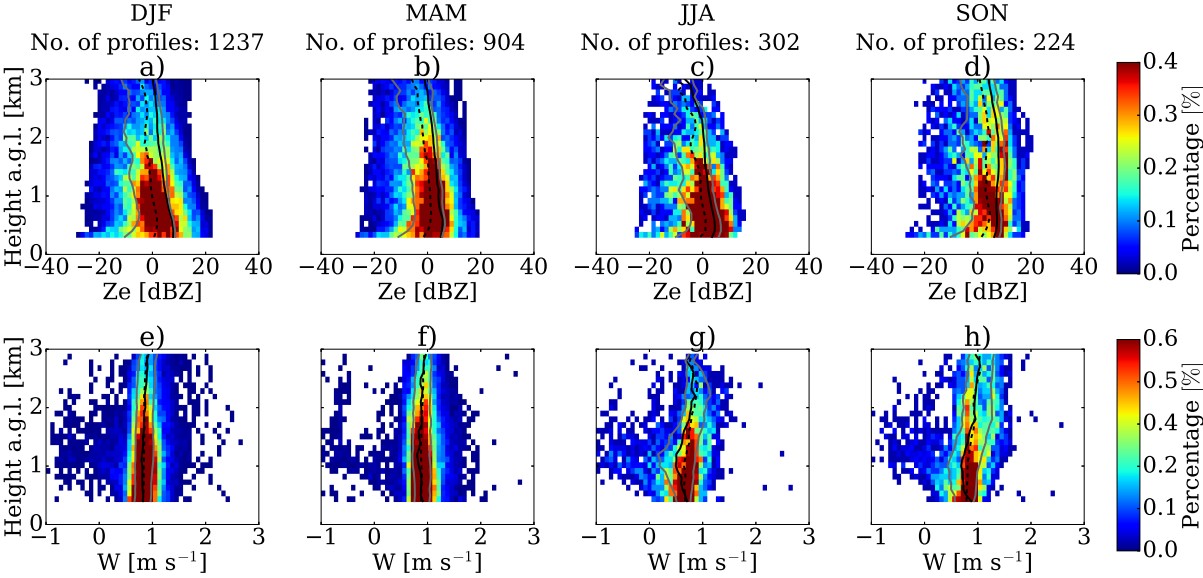

**Figure 8.** Same as Figure 7, but for Princess Elisabeth station.

precipitation is region-dependent, it is fundamental to perform a seasonal analysis of the vertical structure of the precipitation at the two study areas.

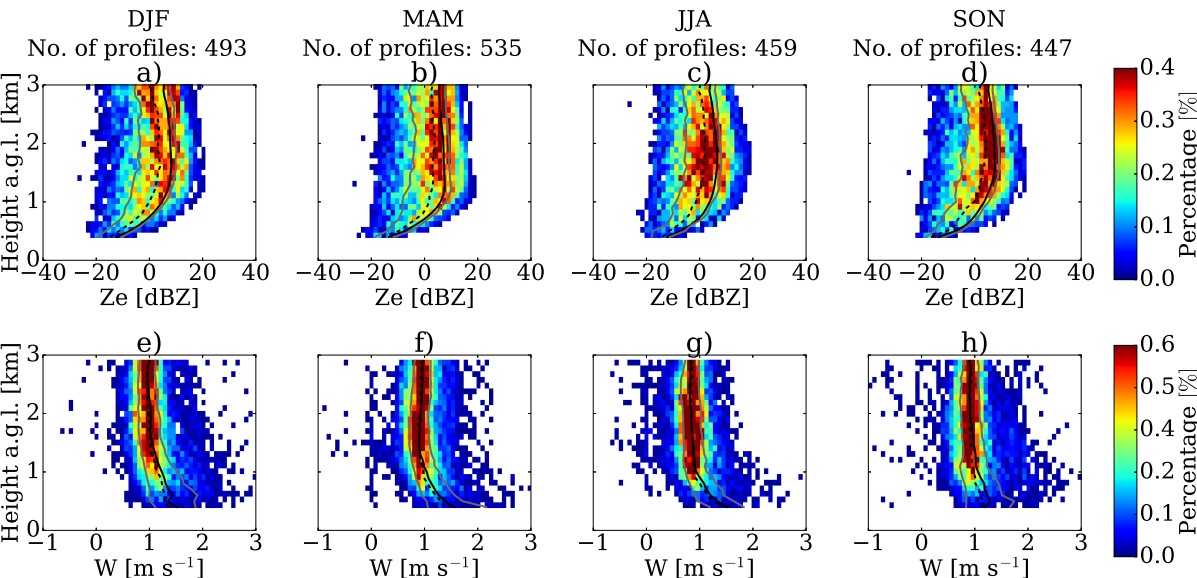

**Figure 9.** Frequency by altitude diagram for Ze (a, b, c) and W (e, f, g) values observed at DDU during DJF (a, e), MAM (b, f), JJA (c, g) and SON (d, h) of virga. Solid-black and dashed-black lines represent the average and median vertical profiles of Ze and W respectively. Grey lines correspond to the 20 and 80% quantiles of the vertical profiles.

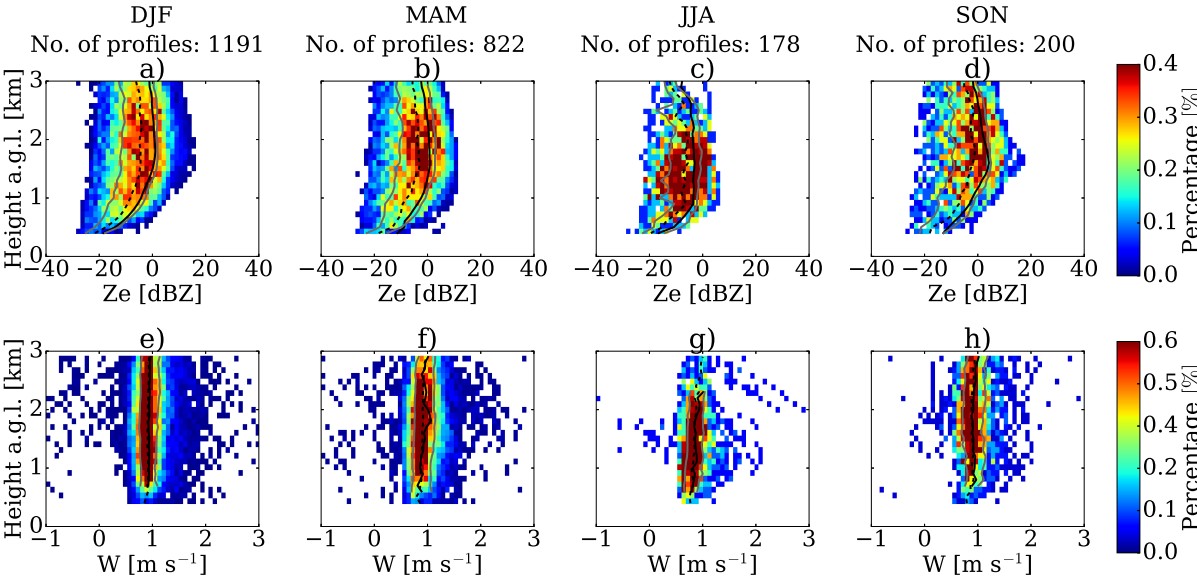

**Figure 10.** Same as Figure 9, but for Princess Elisabeth station.

In general terms, observations are distributed with 21, 27, 25 and 27% for December-January-February (DJF), March-April-May (MAM), June-July-August (JJA) and September-October-November (SON) respectively at DDU, while at PE the

distribution is more concentrated during DJF (48%) and MAM (34%) because of power failures occurred during winter, when the station is not active. Profiles for JJA and SON at PE correspond to the 9 and 8% of the observations, therefore the overall statistics may be biased towards DJF and MAM at PE.

Figure 6 show the average radio soundings of the air temperature ($T$) and the relative humidity with respect to the ice ($RH_i$) for co-located profiles of MRR during surface precipitation and virga events. At DDU, the vertical profiles of $T$ show a seasonal cycle during both types of precipitation. For the cases of $RH_i$, a seasonality is not clearly observed, however profiles corresponding to virga events are systematically drier than those for surface precipitation, conditions expected when solid particles are completely sublimated before reaching the surface. Radio soundings at PE show colder profiles in summer compared to DDU and also drier troposphere during virga events compared to precipitation reaching the ground.

## 5.1 Surface precipitation

During the different seasons at DDU, surface precipitation remains dominant (56 to 69% of occurrence, see Table 1), with the lowest proportion during summer. Figure 7 shows the variability of the VPR and VPV for the different seasons at DDU. Despite that the 3 km range of the radar does not display the full extent of the precipitation systems, this higher percentage of observations in the highest gates suggests that the vertical extension of precipitation events in summer is larger than in the other seasons.

The mean and median VPV at DDU show low variations in the top and then an increase toward the surface. During summer the height at which W starts to increase (2.5 km) is higher compared to autumn and spring (2 km) and winter (1 km). The aggregation process is more favored when temperatures are higher than -15°C (Hosler and Hallgren, 1960; Hobbs et al., 1974), thus a possible explanation for this pattern is that in summer particles have a higher probability to increase in size, since the environmental conditions are favorable for aggregation, according to the radio soundings collected at DDU during precipitation events of the present study (see Figure 6) and previous long term observations (Mygard et al., 2013). The influence of temperature on the vertical radar profiles in the study area, in particular for surface precipitation, is detailed in the Appendix D. On the other hand, the longer path that particles travel during summer increases the probability of reaching bigger size and W.

At PE, surface precipitation is also dominant in all seasons, but with a slightly lower occurrence compared with DDU. The season with lower percentage of surface precipitation is summer (51%), on the other hand, winter presents the higher proportion of surface precipitation (63%). During all seasons, Ze frequencies are highest close to the ground (compared to DDU) and the percentage of observations decreases as the altitude increases (see Figure 8). In winter, the occurrence of precipitation at high altitude levels is lower compared to the other seasons, similarly to DDU.

During all seasons at PE, mean and median VPV show a slight decrease along the vertical profile. Summer and autumn present a more steady decrease, compared with winter and spring, because the latter two decrease until above 1km, then W increase along the lower km. These breaks in the VPV signatures are associated with the reflectivity of shallow precipitation event that are present in all seasons with low frequency, but enough during JJA and SON to impact the shape of the VPV. These

shallow profiles of precipitation are characterized by very low to negative values of W at the top, followed by an increase toward the surface.

## 5.2 Virga

Values of reflectivity factor and vertical velocity at DDU shown in Figures 9 exhibit similar patters in all the seasons, mean VPR decreases rapidly in the lowest 1 km towards the surface. For the cases of VPS at DDU, it shows an increase towards the surface. On the other hand, at PE, the seasonal values of reflectivity also decrease near to the surface, similar to the shape of the vertical profiles at DDU (see Figure 10), however mean and median VPV are relatively constant along the vertical path.

There is a seasonal variability at the two stations, with virga more frequent in summer and autumn compared to winter and spring (see Table 1). The differences in proportion of surface precipitation and virga observed during the season at DDU can be explained due to the seasonal variability of the air temperature. Figure 6a presents the air temperature ($T$) for surface precipitation (solid-lines) and virga (dashed-lines) based on synchronous radio soundings - MRR profiles at DDU. According to Clough and Franks (1991) and Maahn et al. (2014), relatively warm air (as the case of summer) and low relative humidity lead to more sublimation of ice particles. Relative humidity, plays an important role in the sublimation of the particles, but in the case of DDU, relative humidity with respect to ice ($RH_i$) shows no clear seasonality (See Figure 6b). Although during summer the moisture content in the troposphere is higher compared to winter, $RH_i$ decreases as the temperature increases. Figure 6b displays the $RH_i$ for virga and surface precipitation, showing that virga profiles are characterized by less saturated conditions compared with surface precipitation, with a difference in $RH_i$ between 10 and 35%.

## 6 Summary and conclusions

In this study we present a multi-year characterization of the vertical structure of precipitation at two stations in East Antarctica, using vertical profiles of reflectivity (VPR), vertical velocity (VPV) and spectral width (VPS) from micro rain radars. The shape of these vertical profiles evidences the influence of the climatological patterns that impact on precipitation processes at the two stations. The coastal location of DDU provides relatively warmer and moister conditions than at PE, which is located at a higher altitude in the escarpment zone. Analyzing the statistical distribution of the long term observations of the VPR and VPV demonstrated that at DDU there is a higher occurrence of more intense precipitation events with larger vertical extent than at PE. Higher frequencies of large Ze and W values at DDU compared to PE were used as a proxy for the occurrence of intense precipitation. The strong katabatic winds blowing at DDU have a significant impact on the VPR due to a low-level sublimation process. Topographic conditions at PE protect the station from the direct impact of katabatic winds and from strong sublimation of precipitation near the surface.

Two contrasted shapes of VPV are observed in the study areas, influenced by the microphysics of ice particles and the lower tropospheric conditions. Although both stations are characterized by the presence of small ice particles during snowfall events, relatively warmer and moister conditions at DDU favor the occurrence of aggregation and riming of crystals, increasing

the mean vertical velocity toward the surface. On the other hand, the cooler and drier conditions at PE limit the ice-growing processes, leading to a more uniform VPV, with even a decrease towards the surface due to the increase in air density.

The multi-year observations show that virga is a frequent phenomenon in the study areas, corresponding to 36% and 47% of all precipitation profiles at DDU and PE. The vertical profiles of virga are characterized by lower values of radar reflectivity and associated with drier and warmer atmospheric conditions. At both stations, significant differences in the shape of the VPR of surface precipitation were observed when the virga profiles were included. This takes particular importance in the calibration and validation of satellite products for the monitoring of precipitation, because the blind zone limits the differentiation of surface precipitation from virga. Virga appears more frequently in summer, when most of the observation in Antarctica are carried out because of logistical reasons. Winter is the season when virga is less frequent and precipitation events are shallower compared to the other seasons.

The present study explores unique datasets of micro rain radar measurements, which demonstrate a great potential for long-term monitoring of the vertical structure of precipitation in a remote region as Antarctica. Nevertheless, it is necessary to extend this analysis to additional locations across the continent (once large enough datasets become available), in order to improve the characterization of different climatological patterns of precipitation. Moreover, new intensive field campaigns to collect more detailed information on microphysics of hydrometeors also are important to improve the interpretation of the results. Current statistics and future measurements will contribute significantly to better understand the Antarctic precipitation and to evaluate satellite products and verification of numerical precipitation models.

## 7 Appendix A: Variable noise threshold in MK12

The MK12 post-processing (version 0.101) analyses the variance of a given average spectrum to discriminate pure noise from signal that contains a real peak (see equation 5 in MK12). The minimum threshold is:

$$V_T = 0.6/\sqrt{\Delta t}, \tag{4}$$

where $V_T$ is the normalized standard deviation of a single average spectrum and $\Delta t$ is the averaging time. When MRR collects less than the sampling rate, the noise of the signal increases, thus $V_T$ also increase allowing some profiles not to be filtered correctly because the threshold is fixed. A new approach to avoid including noise data in the post-processing is to consider a variable threshold as function of the number of acquisition per minute. A new threshold for the normalized standard deviation was configured as:

$$V_T = 0.6/\sqrt{\Delta t \frac{n}{sr}}, \tag{5}$$

where $n$ is the number of observation per minute, $\Delta t$ is the averaging time equal to 60 seconds and $sr$ is the mean sampling rate equal to 5.7 spectra per minute. This dynamic threshold allows to detect noise that increases its variability due to the decrease of number of observations.

# 8 Appendix B: Equations for saturation vapor pressure over water and ice

To derive the saturation vapor pressure over water $e_{sw}$ and the saturation vapor pressure over ice $e_{si}$ as function of the air temperature, we use the following equations from Goff (1957):

$$log(e_{sw}) = a_1 \cdot \left(1 - \frac{T_0}{T}\right) + a_2 \cdot log\left(\frac{T}{T_0}\right) + a_3 \cdot 10^{-4} \cdot \left(1 - 10^{a_4 \cdot \left(\frac{T}{T_0} - 1\right)}\right) + a_5 \cdot 10^{-3} \cdot (10^{a_6 \cdot \left(1 - \frac{T_0}{T}\right)} - 1) + a_7 \tag{6}$$

and

$$log(e_{si}) = b1 \cdot \left(\frac{T_0}{T} - 1\right) + b_2 \cdot log\left(\frac{T_0}{T}\right) + b_3 \cdot \left(1 - \frac{T}{T_0}\right) + b_4 \tag{7}$$

where $T$ is the air temperature in K, $T_0$ is the triple point of water (273.16 K), $log$ is the logarithm with base 10 and the value for the constants are $a_1$ = 10.79574, $a_2$ = -5.02800, $a_3$ = 1.50475, $a_4$ = -8.2969, $a_5$ = 0.42873, $a_6$ = 4.76955, $a_7$ = -0.21386 $b_1$ = -9.096853, $b_2$ =-3.566506 and $b_3$ = 0.876812. $b_4$ = -0.21386. All the values of pressure are expressed in hPa.

# 9 Appendix C: VPV with respect to the altitude above sea level

To better understand the differences in the VPV at DDU and PE, we compared the VPV at both stations as a function of the height above sea level, instead of the height above ground level. This analysis allows to obtain a common region of altitude where the vertical velocities can be directly compared (see Fig. 11). The left panel of the figure, shows the VPV (mean, median and quantiles) for surface precipitation and the right panel corresponds to virga. In the first case, although at DDU and PE similar values of W are observed at 3km of altitude, VPV at DDU increases rapidly going towards the surface, unlike at PE where a slight decrease is seen. These observed differences go in the same line as the results already discussed in the manuscript. Differences in the dominant microphysical processes (e.g. occurrence of riming and/or aggregation) at both stations play a significant role in the vertical profiles of mean Doppler velocity.

In the second case (virga), VPVs in the common region show very similar pattern for the both stations. An explanation for the similarities observed in the two regions is that the hydrometeors of virga profiles are mainly small size/pristine particles susceptible to be completely sublimated, thus the effect of the air density is similar at DDU and PE in the common region.

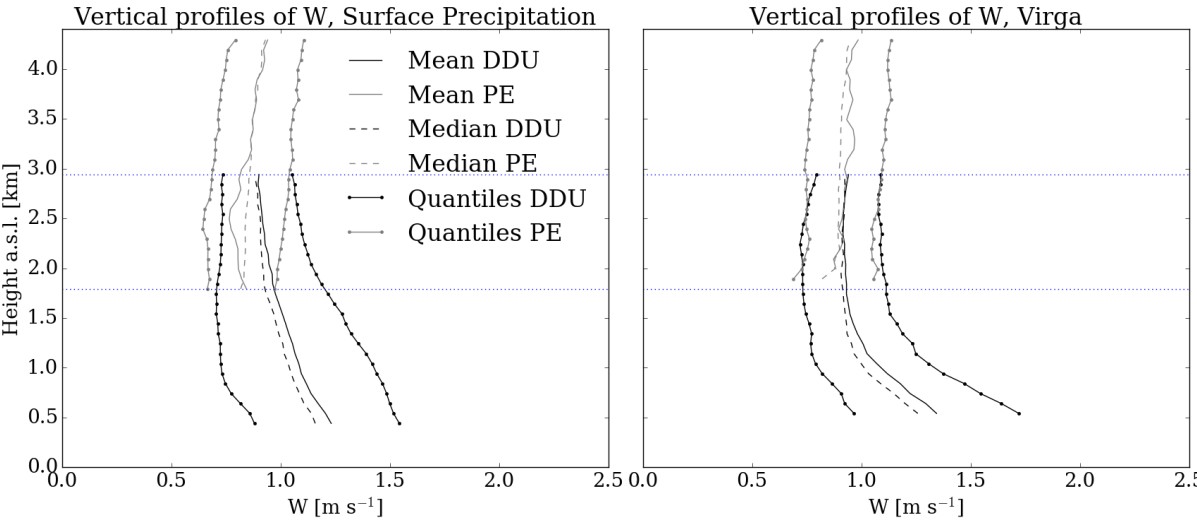

**Figure 11.** Vertical profile of mean Doppler velocity at DDU (black lines) and PE (grey lines). Solid and dashed lines correspond to mean and median profiles, respectively. dotted-solid lines represent the 20 and 80% quantiles. The curves are equivalent to Figure 4e and f and Figure 5e and f, but height is expressed in altitude above mean sea level. Horizontal dotted-blue lines delimit the common height region of the profiles at both stations.

## 10   Appendix D: Temperature and Doppler moments

We have analyzed in more detail VPR as a function of temperature using radio sounding information at DDU and PE. Fig. 12 displays the joint distribution for Ze and temperature, separated by surface precipitation (a) and virga (b). The results show a link between temperature and VPR for surface precipitation profiles at DDU. There is a positive correlation observed in
5   the case of surface precipitation, that may be associated with the efficiency of the particle aggregation process with respect to the temperature. The spread observed for the lower values of Ze, may be linked with particles that are not involved in the aggregation process during surface precipitation.

In the case of virga, this relation is not observed, suggesting that ice particle growth is less significant during this type of events. In the case of PE, the few available radio soundings are not enough to observe a clear relation between both variables.
10   The analysis of VPV and VPS provides similar conclusion (see Fig. 13 and Fig. 14), but the correlation observed for surface precipitation cases is less evident, because these variables, specially spectral width, are affected by the turbulent katabatic winds at DDU.

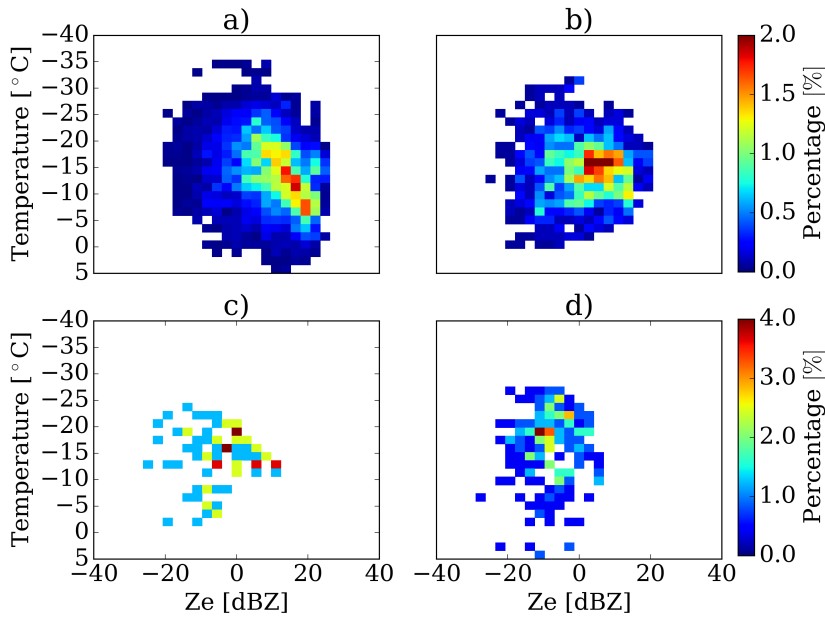

**Figure 12.** Joint distribution for temperature and Ze using radio soundings separated by surface precipitation (a and c) and virga (b and d), for DDU (a and b) and PE (c and d).

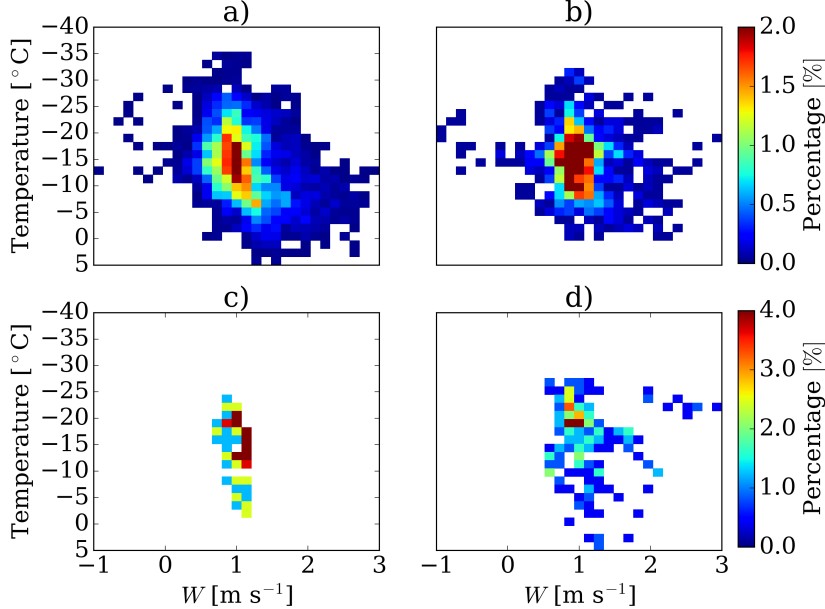

**Figure 13.** Same as Fig. 12, but for W.

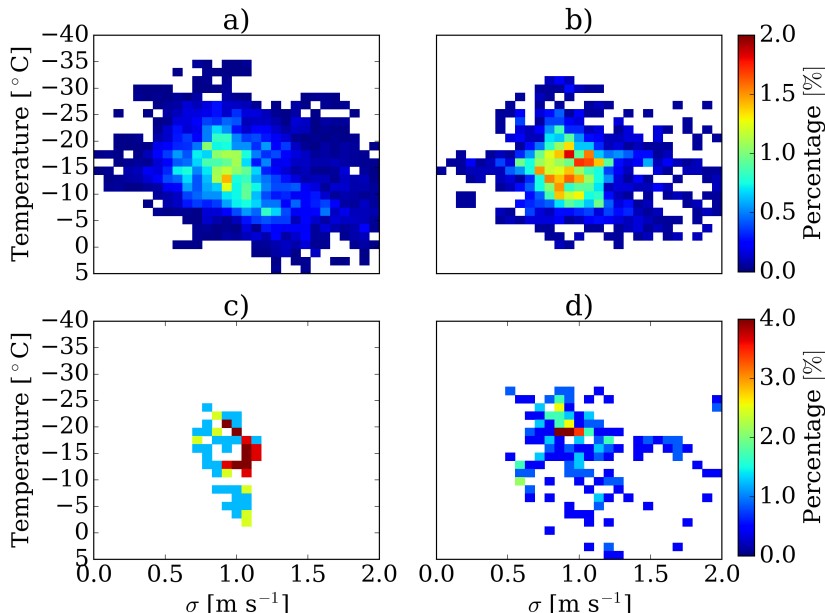

**Figure 14.** Same as Fig. 12, but for $\sigma$.

*Competing interests.* The authors declare that no competing interests are present.

*Acknowledgements.* We thank the French Polar Institute, which logistically supports the APRES3 measurement campaigns at DDU. We thank A. Flouttard, V. Terol, P. Rouault and R. Vérité from Météo-France for taking care of the our AWS at DDU. We thank the logistical support in the maintenance of the instruments at PE. We also acknowledge the support of the French National Research Agency (ANR; grant number: ANR-15-CE01-0003) to the APRES3 project. The project Expecting Earth-Care, Learning from the ATrain (EECLAT) funded by the Centre National d'Etudes Spatiales (CNES) also supported this work. The Swiss National Science foundation SNF is acknowledged for grant 200021_163287, financing the Swiss participation to the project. KUL members and IG thank the support of the Belgian Science Policy Office (BELSPO; grant number: BR/143/A2/AEROCLOUD) and the Research Foundation Flanders (FWO; grant number: G0C2215N). We thanks G. Peters, for the support regarding to MRR operation and post-processing. We thanks the contribution of the anonymous reviewers to the improvement of the present work, in particular for proposing to include appendices C and D. We thank Catherine Wilcox for reviewing the English writing of the manuscript.

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
