# Peer review of "The vertical structure of precipitation at two stations in East Antarctica derived from micro rain radars"

_The Cryosphere, 2018_

## Referee Comment (RC1) · Anonymous Referee #1 · 2 Oct 2018

General comments:

In the manuscript by Durán-Alarcón et al. multi-year micro rain radar observations at two observatories in Antarctica are analyzed. The main focus of the paper lies on the description and interpretation of the statistics of vertical profiles of radar reflectivity, mean Doppler velocity, and spectral width for the two sites. The profiles were separated with respect to occurrence of precipitation on the ground versus virga clouds as well as with respect to different seasons.

Overall, I find this a relevant study of how radar profiles change within the blind zone of most space-borne radar systems. This is very relevant considering snowfall retrievals

and climatologies over Antarctica as well as for providing statistical guidance for numerical models of how precipitation is altered close to the ground. I have some questions and suggestion mainly related to the methodology of the analysis which I described in more detail below. I think extending and clarifying the methods will help to better understand the analysis applied as well as for interpreting the results; it will also help to make the results easily reproducible. After these revisions I can support the paper for publication in The Cryosphere.

Specific Comments: (Page, Line)

6, Section 2.5: I have a couple of questions regarding how the averaging and other statistics are exactly done. I would like to see these methods described in more detail. For example: When you estimate the effect of the different averaging windows, how do you deal with periods when you have no clouds or periods when you have clouds for example at high altitude (2-3km) but clear sky further below (1-2km)? Wouldn't you need a continuous cloud/precipitation system throughout the maximum 12h averaging time in order to have the same sample size at each height within the 3km range? Are you excluding events when the cloud is only lasting for 10h since you cannot average in this case over 12h? Also Equation 2 is not completely clear to me: What is the average VPR (Ze-bar)? Is it the average over all heights or the average of Ze for all cases for a specific height? But then for which integration time? In the end you identify a 1h integration time as optimal but again, how do you deal with profiles which are not continuous over the full 3km range and/or events which are shorter than 1h?

6, 25: I agree that you will probably eliminate variations due to fall streaks with a 1h integration time. But your argumentation with the fall velocity of snowflakes is not correct. The integration over the observation time of the radar is not simply related to the sedimentation time of the hydrometeors. For example, let's imagine there is no horizontal wind shear. This would result in the radar time-height plot in a perfectly vertical fall streak which passes the radar within one sampling time interval (or shorter dependent on advection speed). The fall streak would look exactly the same no matter

how long the particle needs to sediment over a certain altitude range. Quite often, fall streaks are mis-interpreted as trajectories which they usually are not.

7, 5-10: Please specify the Ze-threshold you use to define precipitation detected by MRR. I assume it is the same threshold for both MRRs? Do both systems have similar sensitivities?

8, 5: Again it is not clear to me when do you count a measurement as a valid profile. For example, if you observe precipitation in just two range gates, is this counted as a profile? If yes, would you then normalize height regions with sparse data with the same profile number as regions with high number of observations? Do you have specific reasons why not plotting your data as colored frequency by altitude diagrams (CFADs) where you normalize by the actual number of observations separately for each altitude? I am not sure if this might be superior for your analysis, but I would like to better understand your choice.

10, Eq. (3) and description: Are you actually correcting your Doppler velocity measurements with this air density correction? I think this would be easily possible as you have radio sonde profiles available (as you mainly need pressure and temperature for this correction, also model profiles for example ECMWF reanalysis should work for a first order correction). Since PE is at 1392m a.s.l. I think you should apply this correction for a better comparison of the profiles from the two stations since simply the shift in altitude and hence air density will have a non-negligible effect on the fall velocities. Looking at your plots, this correction would even enhance the differences you find between both stations!

10, 22: I would add something like "For a vertically pointing radar, spectral width…." since spectral width depends in general on the variability of radial velocity of the targets.

12, 5-10: I am not convinced that only sublimation of the smaller ice particles in virga is increasing your Doppler velocity towards the surface for DDU. Certainly, this is one possible scenario. But I would expect the spectra also to become narrower due to

sublimation of the small ice but in your plots I see sigma increasing. In my opinion this could also indicate that riming/aggregation is also present in virga cases at DDU. Skewness could maybe help to investigate this further since in case of size dependent sublimation, one side of the spectrum should decrease faster than the other.

18, 29-32: You might also want to mention here or in the introduction the recent AWARE field campaign organized by ARM/ASR which brought multiple radars and remote sensors to McMurdo station (https://www.arm.gov/research/campaigns/amf2015aware)

17, 4-10 and throughout the text: So far all your plots are a function of altitude above ground. I see the argument that you want to characterize the profiles in this low altitude region since most of this is missed by satellites. It's just a suggestion but since you have the radio sonde profiles, you could also plot the radar variables as function of cloud temperature instead of height. Considering the strong temperature dependence of aggregation and riming, I could imagine that you find some additional effects related to ice microphysics. Some of these plots could maybe just be added as supplemental information.

Technical Corrections/Typos: (Page, Line)

1, 3: Add comma before and after "however"

Throughout the text: I am always wondering whether "radio sounding" or "radiosounding" is more correct. The AMS glossary says it should be "radio sounding".

6, 2: As turbulence usually describes the effect of multiple eddies I think the plural form is unnecessary

8, 17: "ofaggregates"

9, 2: I think plural form of "remain" and "increase" should be used

10, 7: "hydrometeors . . . are represented"

10, 31: "shows"

11, 3: Is virga really a process or rather a phenomenon?

12, 3: add a "the" before "low troposphere"

15, Table 1: "virga with respect to the total number of vertical profiles during"

17, 33: "with respect to ice"

18, 8-10: Complicated and long sentence. Maybe split in two.

19, 8: "as function of"

---

## Referee Comment (RC2) · Anonymous Referee #2 · 7 Nov 2018

This is a very interesting and timely paper presenting vertical profile measurements of reflectivity, Doppler velocity and spectrum width at station in Antarctica. This study is based on a rather uniques dataset, is worth of publishing just because of this. I would also like to compliment the authors for a very clear presentation of the results and interesting findings. I have only two comments that I hope the authors would take into account:

1. Given different climatological regimes at the stations, it would also be interesting to see how VPR, VPV and VPS vary as a function of temperature. So if you could take a nearest sounding or model temperature out and plot radar variable profiles as a

function air temperature.

2. Given different station altitudes, it would also be beneficial that you would correct your Doppler velocity measurements for air density. It should not be too difficult, you could use the standard atmosphere if you cannot get model or sounding data.

---

## Author Comment (AC1) · 19 Dec 2018

Dear Reviewer,

Please find our responses in the supplementary document.

Please also note the supplement to this comment:
https://www.the-cryosphere-discuss.net/tc-2018-153/tc-2018-153-AC1-supplement.pdf
* * *

---

## Author Comment (AC2) · 19 Dec 2018

**Response to Reviewer's Comments:**
**The vertical structure of precipitation at two stations in East Antarctica derived from micro rain radars**

C. Durán-Alarcón, B. Boudevillain, C. Genthon, J. Grazioli, N. Souverijns, N. P. M. van Lipzig, I. V. Gorodetskaya, and A. Berne

With the present supplementary document we provide our responses to the comments of the Reviewer 1 and 2 of the manuscript tc-2018-153, received the day 02 Oct. 2018 and 07 Nov. 2018, respectively.

The comments of the Reviewers are reported in *italic* font. Quotations of the manuscript in its revised or in its original form are reported in blue.

**Anonymous Reviewer 1**

We would like to thank the important contribution of the Reviewer through their comments and suggestions, which in our opinion were very useful to confirm our findings and to improve the quality of the present manuscript.

**General comments**

*In the manuscript by Durán-Alarcón et al. multi-year micro rain radar observations at two observatories in Antarctica are analyzed. The main focus of the paper lies on the description and interpretation of the statistics of vertical profiles of radar reflectivity, mean Doppler velocity, and spectral width for the two sites. The profiles were separated with respect to occurrence of precipitation on the ground versus virga clouds as well as with respect to different seasons.*
*Overall, I find this a relevant study of how radar profiles change within the blind zone of most space-borne radar systems. This is very relevant considering snowfall retrievals and climatologies over Antarctica as well as for providing statistical guidance for numerical models of how precipitation is altered close to the ground. I have some questions and suggestion mainly related to the methodology of the analysis which I described in more detail below. I think extending and clarifying the methods will help to better understand the analysis applied as well as for interpreting the results; it will also help to make the results easily reproducible. After these revisions I can support the paper for publication in The Cryosphere.*

We thank the comments of the Reviewer that highlight precisely the relevance of this study. Below is given a detailed response to the comments and questions about the manuscript.

**Specific Comments: (Page, Line)**

1. ***6, Section 2.5****: I have a couple of questions regarding how the averaging and other statistics are exactly done. I would like to see these methods described in more detail. For example: When you estimate the effect of the different averaging windows, how do you deal with periods when you have no clouds or periods when you have clouds for example at high altitude (2-3km) but clear sky further below (1-2km)? Wouldn't you need a continuous cloud/precipitation*

*system throughout the maximum 12h averaging time in order to have the same sample size at each height within the 3km range? Are you excluding events when the cloud is only lasting for 10h since you cannot average in this case over 12h?*

*Also Equation 2 is not completely clear to me: What is the average VPR (Ze-bar)? Is it the average over all heights or the average of Ze for all cases for a specific height? But then for which integration time? In the end you identify a 1h integration time as optimal but again, how do you deal with profiles which are not continuous over the full 3km range and/or events which are shorter than 1h?*

More details were included in section 2.5 to make clearer the method to compute the average of VPR. In the new version of the manuscript, the average of VPR is described as follow: Average VPRs are calculated in linear units ($mm^6$ $m^{-3}$) as the sum of the 1-min VPRs along fixed windows of $t$ duration, divided by $t$ and then converted in dBZ. In this way, the value of Ze that would correspond to a constant precipitation rate and unchanging particle microphysical properties over a given time interval is obtained. This averaging method allows to conserve the shape of the VPR and also to take into account the effect of the temporal interval duration on Ze, similar to how it will occur with the precipitation rate.

Based on this definition of the averaging method, the presence of clear sky at different levels of the profiles within a given temporal interval does not represent a problem, which extends for events that last less than the respective time interval. An example could be an event that has a duration of 10-h, integrated in a 12-h interval.

The notation $\overline{Ze}$ was confusing and it was changed to $Ze$ because it only represents the averaged VPR (for the full data period) at given temporal integration. As $Ze$ is computed for the full dataset, $Ze$ is always continuous from the lowest level up to 3km range, thus it is not a problem for the analysis of variability between temporal intervals.

Regarding to the average of VPV and VPS, the last paragraph of the section 2.5 was modified to clarify the method of integration. The new paragraph is the following: After the selection of the temporal integration, VPV and VPS are averaged at the same temporal resolution. In these cases, the average does not considered the zero-values (clear sky situations), as the case of VPR, because $W$ and $\sigma$ are considered as intensive properties of precipitation, which means that they are independent of the amount of precipitation within a given temporal interval.

2. ***6, 25****: I agree that you will probably eliminate variations due to fall streaks with a 1h integration time. But your argumentation with the fall velocity of snowflakes is not correct. The integration over the observation time of the radar*

*is not simply related to the sedimentation time of the hydrometeors. For exam-*
*ple, let's imagine there is no horizontal wind shear. This would result in the*
*radar time-height plot in a perfectly vertical fall streak which passes the radar*
*within one sampling time interval (or shorter dependent on advection speed).*
*The fall streak would look exactly the same no matter how long the particle*
*needs to sediment over a certain altitude range. Quite often, fall streaks are*
*mis-interpreted as trajectories which they usually are not.*

Thank you very much for this comment. Indeed, the temporal integration of the
vertical profiles will not necessarily remove the fall streaks, but it will smooth them
and reduce the variability in the dataset. As Figure 2 of the paper shows, after
1-h integration, the variability of all precipitation profiles does not show significant
changes, thus 1-h is chosen as references of the temporal integration.
The sentence in page 6 line 25 is not right. Wind shear and horizontal wind advection
are also important factor controlling fall streaks, and not only the sedimentation
velocity. This sentence was removed in the last version of the manuscript, because
it was not a correct argument for the choice of integration time. On the other hand,
Figure 2 is sufficient for the justification of the selected integration time.

3. **7, 5-10**: *Please specify the Ze-threshold you use to define precipitation detected*
   *by MRR. I assume it is the same threshold for both MRRs? Do both systems*
   *have similar sensitivities?*

Both instruments have the same sensitivity (between -14 and -8 dBZ from the lowers
available level (300m) to the top (3km), Maahn and Kollias (2012)), however, as the
MRR located at DDU is protected with a radome, its effective sensitivity is reduced
in about 6 dBZ according to the report of Grazioli et al. (2017a). This information
was clarified in the new version of the manuscript in section 2.3.
The new sentences are the following: After post-processing, the sensitivity of MRR
ranges betwen -14 and -8 dBZ depending of the height level (Maahn and Kollias,
2012). At DDU, the attenuation due to the radome (estimated at about 6 dBZ, see
Grazioli et al., (2017a)) must however be taken into account and leads to a lower
sensitivity.

4. **8, 5**: *Again it is not clear to me when do you count a measurement as a valid*
   *profile. For example, if you observe precipitation in just two range gates, is this*
   *counted as a profile? If yes, would you then normalize height regions with sparse*
   *data with the same profile number as regions with high number of observations?*

*Do you have specific reasons why not plotting your data as colored frequency by altitude diagrams (CFADs) where you normalize by the actual number of observations separately for each altitude? I am not sure if this might be superior for your analysis, but I would like to better understand your choice.*

After removing the noise and artifacts, all the profiles were included in the analysis. The new version of the manuscript was modified, because the counts are normalized by the total number of observed precipitation gates and not by the total number of profiles. This choice avoids problems with altitude ranges that have different number of precipitation profiles and unlike a CFAD, these figures also provide information about the vertical distribution of the observations.

5. ***10, Eq. (3) and description***: *Are you actually correcting your Doppler velocity measurements with this air density correction? I think this would be easily possible as you have radio sonde profiles available (as you mainly need pressure and temperature for this correction, also model profiles for example ECMWF reanalysis should work for a first order correction). Since PE is at 1392m a.s.l. I think you should apply this correction for a better comparison of the profiles from the two stations since simply the shift in altitude and hence air density will have a non-negligible effect on the fall velocities. Looking at your plots, this correction would even enhance the differences you find between both stations!*

Mean Doppler velocity values were not corrected from the effect of air density. The Eq. 3 was cited to explain the different patterns observed in the VPV at the two sites. We consider that this correction may mislead the main objective of this part of the analysis, which is to characterize the profiles of actual vertical velocity. In addition, even after the air density correction, other variables, such as temperature or humidity, that determine riming/aggregation processes, would still be affecting the vertical pattern of W in different ways at both stations. To better understand the differences in the VPV at DDU and PE, we propose to compare the profiles of W at both stations as function of the height above sea level, instead of the height above ground level. This analysis allows to obtain a common region of altitude where the vertical velocities can be compared (see Fig. I). The left panel of the figure, shows the VPV (mean, median and quantiles) for surface precipitation and the right panel corresponds to virga. In the first case, although at DDU and PE similar values of W are observed at 3km of altitude, VPV at DDU increases rapidly going towards the surface, unlike PE which shows a slight decrease. These observed differences go in the same line as the results already discussed in the manuscript. Differences in the

[Figure]

Figure I: Vertical profile of mean Doppler velocity at DDU (black lines) and PE (grey lines). Solid and dashed lines corresponds to mean and median profiles, respectively. dotted-solid lines represent the 20 and 80% quantiles. The curves are equivalent to Figure 4e and f and Figure 5e and f (in the manuscript), but height is expressed in altitude above mean sea level. Horizontal dotted-blue lines delimit the common height region of the profiles at both stations.

dominant microphysical processes (e.g. occurrence of riming and/or aggregation) at both stations play a significant role in the vertical profiles of mean Doppler velocity. In the second case (virga), VPVs in the common region show very similar pattern at both stations. This is an interesting result that was not highlighted in the first version of the manuscript. An explanation for the similarities observed in the two regions is that the hydrometeors of virga profiles are mainly small size/pristine particles susceptible to be completely sublimated, thus the effect of the air density is similar at DDU and PE in the common region.

Figure I was included in the revised version of the manuscript, as an Appendix, to complement the interpretation of the results of the section 3.2.

6. **10, 22**: *I would add something like "For a vertically pointing radar, spectral width…." since spectral width depends in general on the variability of radial velocity of the targets.*

Indeed, the phrase refers exclusively to the spectral width derived from vertically pointing radar. The suggestion of the Reviewer was included in the new version.

7. **12, 5-10**: *I am not convinced that only sublimation of the smaller ice particles in virga is increasing your Doppler velocity towards the surface for DDU.*

[Figure]

Figure II: Joint distribution of wind direction and spectral width for surface precipitation (a) and virga cases at DDU, based on co-located radio soundings and MRR observations.

> *Certainly, this is one possible scenario. But I would expect the spectra also to become narrower due to sublimation of the small ice but in your plots I see sigma increasing. In my opinion this could also indicate that riming/aggregation is also present in virga cases at DDU. Skewness could maybe help to investigate this further since in case of size dependent sublimation, one side of the spectrum should decrease faster than the other.*

We agree with the Reviewer that riming/aggregation may play a role in the increase of the mean Doppler velocity towards the surface. However, if these factors are present, they have a second order influence. This is evidenced by the strong decrease in the frequency of precipitation gates observed towards the surface (Fig. 4f in the manuscript), as well as by the significant decrease of Ze (Fig. 4c in the manuscript). If riming/aggregation had a primary role in the increase of W, Ze should increase as well.
With respect to the increase of spectral width in the lowest kilometer, the turbulence of the strong katabatic layer at DDU plays an important role. Due to the characteristic directionality of the katabatic winds at DDU from the southeast, in Fig. II it is possible to appreciate its influence on the increase of spectral width.
The paragraph was complemented with the following sentence: The occurrence of riming/aggregation of the hydrometeors may also play a secondary role in the increase of W toward the surface observed at DDU in virga profiles.

8. ***18, 29-32****: You might also want to mention here or in the introduction the*

*recent AWARE field campaign organized by ARM/ASR which brought multiple radars and remote sensors to McMurdo station (https://www.arm.gov/research/ campaigns/amf2015aware)*

The project is mentioned in the introduction of the new version of the manuscript.

9. ***17, 4-10 and throughout the text****: So far all your plots are a function of altitude above ground. I see the argument that you want to characterize the profiles in this low altitude region since most of this is missed by satellites. It's just a suggestion but since you have the radio sonde profiles, you could also plot the radar variables as function of cloud temperature instead of height. Considering the strong temperature dependence of aggregation and riming, I could imagine that you find some additional effects related to ice microphysics. Some of these plots could maybe just be added as supplemental information.*

Thank you very much for this suggestion. We have analyzed in more detail the radar variables using radio sounding information at DDU and PE. The results show an interesting link between temperature and Ze for surface precipitation profiles at DDU. Fig. III displays the joint distribution for Ze and temperature, separated in surface precipitation (a) and virga (b). There is a positive correlation observed in the case of surface precipitation, that may be associated with the efficiency of the particle aggregation process with respect to the temperature. The spread observed for the lower values of Ze, may be linked with particles that are not involved in the aggregation process in surface precipitation profiles.

In the case of virga, this relation is not observed, suggesting that ice particle growth is less significant during this type of events. In the case of PE, the few available radio soundings are not enough to observe a clear relation between both variables.

We include in the manuscript the Figure III to show in more detail the effect of temperature in the vertical structure of the precipitation.

**Technical Corrections/Typos: (Page, Line)**

1. ***1, 3****: Add comma before and after "however"*

The comma was added.

[Figure]

Figure III: Joint distribution for temperature and Ze using radio soundings separated by surface precipitation (a and c) and virga (b and d), for DDU (a and b) and PE (c and d).

2. ***Throughout the text****: I am always wondering whether "radio sounding" or "radiosounding" is more correct. The AMS glossary says it should be "radio sounding".*

Thank you for this correction, the word was modified.

3. ***6, 2****: As turbulence usually describes the effect of multiple eddies I think the plural form is unnecessary*

Turbulence is written in singular in the new version of the manuscript.

4. ***8, 17****: "ofaggregates"*

Corrected.

5. ***9, 2****: I think plural form of "remain" and "increase" should be used*

You are right, the text was modified.

6. **10, 7**: *"hydrometeors ... are represented"*

Corrected.

7. **10, 31**: *"shows"*

Corrected.

8. **11, 3**: *Is virga really a process or rather a phenomenon?*

Indeed, virga is an atmospheric phenomenon. The manuscript was modified.

9. **12, 3**: *add a "the" before "low troposphere"*

Added.

10. **15, Table 1**: *"virga with respect to the total number of vertical profiles during"*

Thank you for the correction, the missing words were added in the caption of the Table 1.

11. **17, 33**: *"with respect to ice"*

Text modified.

12. **18, 8-10**: *Complicated and long sentence. Maybe split in two.*

The sentence was modified as follows: The coastal location of DDU provides relatively warmer and moister conditions than at PE, which is located at a higher altitude in the escarpment zone. Our results demonstrated that at DDU there is a higher occurrence of more intense precipitation events with larger vertical extent compared with PE, by analyzing the statistical distribution of the long term observations of the VPR and VPV. Higher frequencies of large Ze and W values at DDU compared to PE, were used as a poxy for the ocurrence of intense precipitation.

13. **19, 8**: *"as function of"*

The sentence was modified.

**Anonymous Reviewer 2**

We thank Reviewer 2 for their positive comments. They helped us analyze in more detail the impact of the hydrometeor microphysics on vertical structure of precipitation.

*This is a very interesting and timely paper presenting vertical profile measurements of reflectivity, Doppler velocity and spectrum width at station in Antarctica. This study is based on a rather uniques dataset, is worth of publishing just because of this. I would also like to compliment the authors for a very clear presentation of the results and interesting findings. I have only two comments that I hope the authors would take into account:*

1. *Given different climatological regimes at the stations, it would also be interesting to see how VPR, VPV and VPS vary as a function of temperature. So if you could take a nearest sounding or model temperature out and plot radar variable profiles as a function air temperature.*

Thank you very much for this suggestion. We have analyzed in more detail VPR as function of temperature using radio sounding information at DDU and PE. Fig. IV displays the joint distribution for Ze and temperature, separated by surface precipitation (a) and virga (b). The results show an interesting link between temperature and VPR for surface precipitation profiles at DDU. There is a positive correlation observed in the case of surface precipitation, that may be associated with the efficiency of the particle aggregation process with respect to the temperature. The spread observed for the lower values of Ze, may be linked with particles that are not involved in the aggregation process in surface precipitation profiles.
In the case of virga, this relation is not observed, suggesting that ice growth is less significant during this type of events.
In the case of PE, the few available radio soundings are not enough to observe a clear relation between both variables.
The analysis of VPV and VPS provides similar conclusion (please see Fig. V and Fig. VI), but the correlation observed for surface precipitation cases is less evident, because these variables, specially spectral width, are affected by the turbulent katabatic winds at DDU.
We complemented the manuscript with the Figures IV, V and VI to show in more detail the effect of temperature in the vertical structure of the precipitation.

[Figure]

Figure IV: Joint distribution for temperature and Ze using radio soundings separated by surface precipitation (a and c) and virga (b and d), for DDU (a and b) and PE (c and d).

[Figure]

Figure V: Same as Fig. IV, but for W.

[Figure]

Figure VI: Same as Fig. IV, but for $\sigma$.

2. *Given different station altitudes, it would also be beneficial that you would correct your Doppler velocity measurements for air density. It should not be too difficult, you could use the standard atmosphere if you cannot get model or sounding data.*

We think that such a correction could be misleading with respect to the main objective of this part of the analysis, which is to characterize the profiles of actual vertical velocity. In addition, even after the air density correction, other variables, such as temperature or humidity, that condition the occurrence of riming/aggregation processes, would still be affecting the vertical pattern of W in different ways at both stations. To better understand the differences in the VPV at DDU and PE, we propose to compare the profiles of W at both stations as function of the height above sea level, instead of the height above ground level. This analysis allows to obtain a common region of altitude where the vertical velocities can be compared (see Fig. I). The left panel of the figure, shows the VPV (mean, median and quantiles) for surface precipitation and the right panel corresponds to virga. In the first case, although at DDU and PE similar values of W are observed at 3km of altitude, VPV at DDU increases rapidly going towards the surface, unlike PE which shows a slight decrease. These observed differences go in the same line with the results already discussed in the manuscript. Differences in the dominant microphysical processes (e.g. occur-

[Figure]

Figure VII: Vertical profile of mean Doppler velocity at DDU (black lines) and PE (grey lines). Solid and dashed lines corresponds to mean and median profiles, respectively. dotted-solid lines represent the 20 and 80% quantiles. The curves are equivalent to Figure 4e an f and Figure 5e and f (in the manuscript), but height is expressed in altitude above mean sea level. Horizontal dotted-blue lines delimit the common height region of the profiles at both stations.

rence of riming and/or aggregation) at both stations play a significant role in the vertical profiles of mean Doppler velocity.

In the second case (virga), VPVs in the common region show very similar pattern at both stations. This is an interesting result that was not highlighted in the first version of the manuscript. An explanation for the similarities observed in the two regions is that the hydrometeors of virga profiles are mainly small size/pristine particles susceptible to be completely sublimated, thus the effect of the air density is similar at DDU and PE in the common region.

Figure VII was included in the new version of the manuscript, as an Appendix, to complement the interpretation of the results of the section 3.2.

[revised manuscript text omitted]